



# Influence of wind strength and direction on diffusive methane fluxes and atmospheric methane concentrations above the North Sea

Ingeborg Bussmann[1], Eric P. Achterberg[2], Holger Brix[3], Nicolas Brüggemann[4], Götz Flöser[3], Claudia Schütze[5], Philipp Fischer[2]

[1]Departments of Shelf Sea System Ecology & Coastal Ecology, Alfred-Wegener-Institut, Helmholtz Zentrum für Polar- und Meeresforschung, Kurpromenade 201, 27498 Helgoland, Germany

[2]GEOMAR, Helmholtz Centre for Ocean Research, Wischhofstr. 1-3, 24148 Kiel, Germany

[3]Helmholtz-Zentrum Hereon, Institute of Carbon Cycles, Max-Planck-Straße 1, 21502 Geesthacht, Germany

[4]Institute of Bio- and Geosciences – Agrosphere (IBG-3), Forschungszentrum Jülich GmbH, Wilhelm-Johnen-Str., 52425 Jülich, Germany

[5]Department of Monitoring and Exploration Technologies, Helmholtz Centre for Environmental Research -
UFZ, Permoserstr. 15, 04318 Leipzig, Germany

*correspondence to ingeborg.bussmann@awi.de*

**Abstract.** Quantification of the diffusive methane fluxes between the coastal ocean and atmosphere is important to constrain the atmospheric methane budget. The determination of the fluxes in coastal waters is characterized by a high level of uncertainty. To improve the accuracy of the estimation of coastal methane fluxes, high temporal and spatial sampling frequencies of dissolved methane in seawater are required as well as the quantification of atmospheric methane concentrations, wind speed and wind direction above the ocean. In most
cases, these atmospheric data are obtained from land-based atmospheric and meteorological monitoring stations in the vicinity of the coastal ocean methane observations.

In this study, we measured wind speed and direction as well as atmospheric methane directly on board three research vessels in the southern North Sea and compared the local and remote atmospheric and meteorological measurements on the quality of the flux data. In addition, we assessed the source of the atmospheric methane
measured in the study area in the German Bight using airmass back trajectory assessments.

The choice of the wind speed data source had a strong impact on the flux calculations. Fluxes based on wind data from nearby weather stations amounted to only $58 \pm 34\%$ of values based on situ data. Using in-situ data, we calculated an average diffusive methane sea-to-air flux of $221 \pm 351$ µmol m$^{-2}$ d$^{-1}$ (n = 941) and $159 \pm 444$ µmol m$^{-2}$ d$^{-1}$ (n = 3028) for our study area in September 2019 and 2020, respectively. The area-weighted
diffusive flux for the entire area of Helgoland Bay ($3.78 \times 10^9$ m$^2$) was $836 \pm 97$ and $600 \pm 111$ kmol d$^{-1}$ for September 2019 and 2020, respectively. Using the median value of the diffusive fluxes for these extrapolations resulted in much lower values, compared to area-weighted extrapolations or mean-based extrapolations.



In general, at high wind speeds, the surface water turbulence is enhanced and the diffusive flux increases. This enhanced methane input however is quickly diluted within the air mass. Hence, a significant correlation between the methane flux and the atmospheric concentration was observed only at wind speeds < 5 m s$^{-1}$.

The atmospheric methane concentration was mainly influenced by the wind direction, i.e., the origin of the transported air mass. Airmasses coming from industrial regions resulted in elevated atmospheric methane concentrations, while airmasses coming from the North Sea transported reduce methane levels. With our detailed study on the spatial distribution of methane fluxes we were able to provide a detailed and more realistic estimation of coastal methane fluxes.

# 1    Introduction

## 1.1    Necessity for coastal methane data

Methane ($CH_4$) is the second-most important greenhouse gas (GHG) after carbon dioxide ($CO_2$), accounting for 16–25% of atmospheric warming to date (Etminan et al., 2016). Aquatic ecosystems contribute 41% (median) or 53% (mean) of total global $CH_4$ emissions from anthropogenic and natural sources (Rosentreter et al., 2021a). Coastal seas are an important global source of GHGs (Saunois et al., 2020). For the open and coastal ocean including estuaries, Saunois et al. (2020) suggested an emission of 6 (range 2–10) Tg $CH_4$ yr$^{-1}$. A more recent study from Rosentreter suggested an emission of 8.4 (4.8–28.4, Q1-Q3) Tg $CH_4$ yr$^{-1}$, with a contribution of 3% from estuaries, 13% from tidal flats and 52% from continental shelves (Rosentreter et al., 2021a). The near-shore environments hence contribute the largest but most uncertain diffusive fluxes despite accounting for only ~3% of the global ocean area.

The reasons for the large range and uncertainty of coastal $CH_4$ fluxes are associated with the high spatial and temporal variability of fluxes in coastal ecosystems, driven by, for example, variations in tidal pumping and salinity gradients (Rosentreter et al., 2021a), exacerbated by a paucity of data with sufficient temporal and spatial resolution (Weber et al., 2019). Overall, aquatic GHG emissions are causing considerable uncertainty in global GHG assessments (IPCC, 2021). Thus, reducing the uncertainty in aquatic GHG budgets is important to allow improvements to biogeochemical models and climate predictions.

## 1.2    Traditional method for flux calculation

The air–sea gas flux is a function of the gas transfer velocity (k) and atmospheric and oceanic $CH_4$ concentrations (Wanninkhof, 2014, details see method section). Since *k* is difficult to measure, it is often parameterized using widely measured parameters such as wind speed. In offshore regions with greater water depth, wind is known as a good predictor for the gas transfer velocity because wind creates waves and currents which control turbulence and bubbles at the sea surface (Wanninkhof et al., 2009). Also in shallow waters, *k* can be well estimated by wind speed when the water depth is more than 10 m (Ho et al., 2018). Other techniques to determine *k* are eddy covariance measurements, tracer injection methods (Gutiérrez-Loza et al., 2022; Dobashi and Ho, 2023) and chamber measurements (Rosentreter et al., 2021b). The best way to determine *k* is an ongoing matter of debate.

Diffusive $CH_4$ fluxes are typically determined from direct surface ocean $CH_4$ observations and parametrizations of wind speed and atmospheric $CH_4$ concentrations. The atmospheric data used are normally taken from coastal





meteorological stations in close proximity to the marine observations (see for example (Myllykangas et al., 2020; Woszczyk and Schubert, 2021), or a combination of in-situ data and data obtained from a meteorological station is used (Mau et al., 2015; Bussmann et al., 2021b; Humborg et al., 2019). Other studies use in-situ data for all variables (de Groot et al., 2023: Thornton, 2016 #2655). We are not aware of any study on the influence of the data source on the quantification of diffusive CH₄ fluxes.

### 1.3 Atmospheric methane above a water body

The atmospheric CH₄ concentration is determined by several factors. One is the sea-to-air transfer through the diffusive CH₄ flux (Wanninkhof, 2014), implying that periods or areas with high diffusive CH₄ fluxes into the atmosphere would result in higher atmospheric CH₄ concentrations. However, there are contrasting reports in literature in marine science, with highest atmospheric CH₄ concentrations being observed during cruises with lowest CH₄ fluxes (Silyakova et al., 2020). Increasing atmospheric CH₄ levels were not found alongside enhanced dissolved CH₄ concentrations (Vogt et al., 2023; Law et al., 2010). These studies show that there is no clear mechanistic understanding of the relationship between dissolved CH₄ concentrations, CH₄ fluxes to the atmosphere and atmospheric CH₄ concentrations in shallow coastal water areas.

### 1.4 Methane in the North Sea

The CH₄ budget of the central North Sea is characterized by pockmarks (Römer et al., 2021), drilling activities (Vielstädte et al., 2017), and gas ebullition sites (Mau et al., 2015). In contrast, in the southern North Sea and areas close to the mainland, dissolved CH₄ mainly originates from autochthonous methanogenesis in sediments (Yin et al., 2019) with subsequent fluxes into the water column, and also from tidal flats ((Røy et al., 2008; Wu et al., 2015) and riverine inputs (Upstill-Goddard and Barnes, 2016). Borges et al. (2017) showed that warm summers in northern Europe in recent years have resulted in increased dissolved CH₄ concentrations due to enhanced methanogenesis, which has led to higher sea-to-air CH₄ fluxes along the Belgian coast (Borges et al., 2017; Borges et al., 2019).

Previous studies have investigated the temporal and spatial patterns of dissolved CH₄ between the German North Sea coast and the island of Helgoland (60 km offshore) on a monthly basis from 2010 to 2014 (Matousu et al., 2017; Osudar et al., 2015; Hackbusch et al., 2019). In these studies, the CH₄ concentrations near the coast ranged between 30 and 51 nmol L⁻¹, whereas near Helgoland, the concentrations were 14 ± 6 nmol L⁻¹. However, no flux data were calculated in these studies. At these high concentrations of dissolved CH₄ in the coastal North Sea (the equilibrium concentration is 2–3 nmol L⁻¹), the diffusive flux is mainly directed from the sea into the atmosphere.

### 1.5 Aim of study

The aim of this study was to establish the diffusive CH₄ fluxes from the sea into the atmosphere in the southern German Bight (North Sea) based on CH₄ concentration data of high spatial and temporal resolution. We also investigated the influence of the use of different auxiliary data sets on the calculation of the diffusive CH₄ fluxes over a wide area of the Helgoland Bay. We assessed whether increased diffusive CH4 fluxes lead to detectable increases in atmospheric CH4 concentrations and identified the atmospheric factors that influence CH4 concentrations in a given area.



## 2    Material & Methods

### 2.1    Study Sites

The cruises Stern-3 and 5 were performed in September 2019 and 2020, as part of the "Modular Observation Solutions for Earth Systems" (MOSES, (Weber et al., 2021)) subproject "Hydrological Extremes".

On Stern-3, the research vessels RV Littorina (German Helmholtz Centre GEOMAR), RV Ludwig Prandtl (German Helmholtz Centre Hereon) and RV Uthörn (German Helmholtz Centre AWI) left the harbor of Cuxhaven (Fig. 1) on 9 September 2019, heading for the island of Helgoland (German Bight) following different cruise tracks. On 10 September 2019, RV Littorina and Ludwig Prandtl returned to Cuxhaven and the Elbe Estuary, while RV Uthörn returned via the river Weser to Bremerhaven (Bussmann et al., 2020).

For Stern-5, the three research vessels started from Cuxhaven and the Elbe estuary on 30 August 2020, again heading for Helgoland (Fig. 2). In the following days, RV Littorina covered the area between Helgoland and Büsum (on the mainland), extending the cruise track towards the East. RV Ludwig Prandtl covered the area further north and west of the island of Amrum. Mya II (German Helmholtz Centre AWI) covered the area between Helgoland and Bremerhaven with the Weser Estuary. On the last day (3 September 2020) RV Mya II ended the cruise in Sylt, while the others returned to Cuxhaven (Bussmann et al., 2021a).





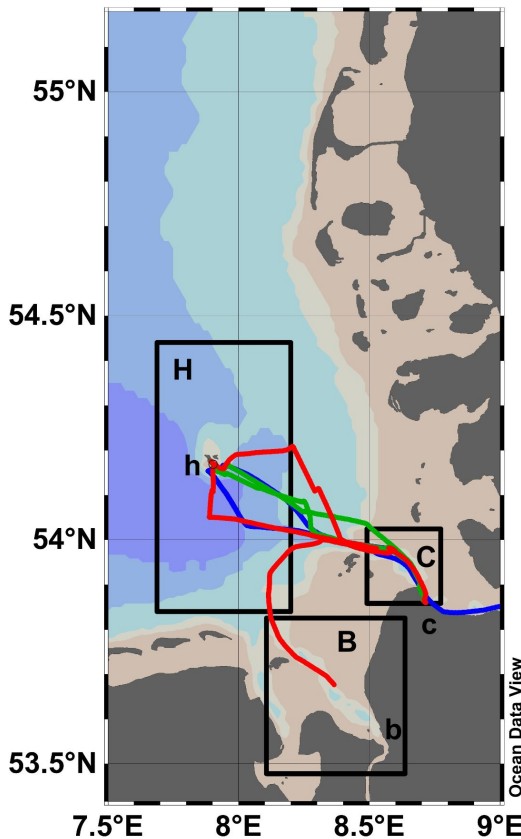

**Figure 1. Cruise tracks for Stern 3, in September 2019 with RVs Littorina (blue), Ludwig Prandtl (green) and Uthörn (red). The areas for different flux calculations are also indicated: the area around Bremerhaven (B) with its meteorological station (b), the area around Cuxhaven (C) with its meteorological station (c) and the area around Helgoland (H) with its meteorological station (h).**



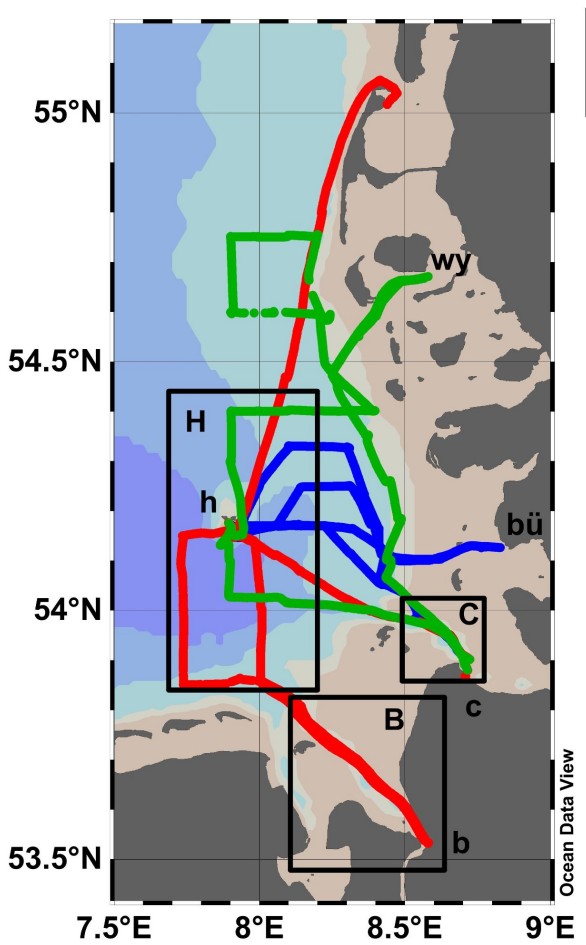

**Figure 2. Cruise tracks for Stern 5, in September 2020 with RVs Littorina (blue), Ludwig Prandtl (green) and Mya II (red). The areas for different flux calculations are also indicated: the area around Bremerhaven (B) with its meteorological station (b), the area around Cuxhaven (C) with its meteorological station (c) and the area around Helgoland (H) with its meteorological station (h).**

## 2.2 Hydrographic and meteorological parameters

Basic hydrographic parameters, such as temperature, salinity, pH, oxygen, turbidity and chlorophyll fluorescence, were measured by shipboard measurement systems (FerryBoxes; 4Hjena, Germany) on all ships. FerryBox systems had been checked and calibrated during routine maintenance and in preparation for the cruises (Petersen, 2014). The water supply to the FerryBoxes and $CH_4$ analyzers (see below section Methane analysis) was taken from the ships' underway surface water supply (intake at 1–2 m). The water from the ship was either pumped through a flow-through basin with ambient air pressure (Stern-3) or from a specific in-situ pump tower (Stern-5) with a 10-l volume, pressure regulator and water overflow. Full details are in the cruise



reports (Bussmann et al., 2020; Bussmann et al., 2021a). Both systems ensured a constant and sufficient surface water supply to all sensors.

True wind speed and true wind direction were provided by the dship-system (nautical data system, Werum) or equivalent systems of Ludwig Prandtl, Mya and Uthörn (no data were available from RV Littorina) with a frequency of 1 min$^{-1}$. Available wind speed data were corrected to $U_{10}$ (at 10 m height) with the respective measuring height:

$$U_{10} = U_{ship} \, (10 \, / \, z_{ship})^{0.143} \tag{1}$$


For any further calculations, we used the rolling mean over 10 minutes. For comparison, we also used wind data (hourly means) provided by the German Meteorological Service (https://cdc.dwd.de/portal/) for the weather stations Bremerhaven, Cuxhaven and Helgoland Dune. For each station, an area was set for which the respective wind data were used for the flux calculations (Fig. 1 and 2). Data on local tidal cycles were provided by the

Bundesamt für Seeschifffahrt und Hydrographie (Federal Maritime and Hydrographic Agency, https://www.bsh.de/DE/DATEN/Vorhersagen/Gezeiten/gezeiten_node.html) for the sites Büsum and Wyk auf Föhr.

### 2.3 Methane Analysis

Dissolved $CH_4$ concentrations were measured with a dissolved gas extraction unit and a laser based analytical Greenhouse Gas Analyzer (GGA; both Los Gatos Research, United States) on all three ships. The degassing devices withdrew water from the flow-through units at 1.2 L min$^{-1}$. Methane was extracted from the water via a hydrophobic membrane and hydrocarbon-free carrier gas on the other side of the membrane (synthetic air or nitrogen, at 0.5 L min$^{-1}$). The carrier gas with the extracted $CH_4$ was then directed to the inlet of the gas

analyzer. The time offset between the water intake and stable recording at the GGA was determined beforehand in the laboratory.

To convert the relative concentrations (ppm) given by the GGA to absolute concentrations (nmol L$^{-1}$), discrete water samples were obtained at least every hour. The $CH_4$ concentration in these bottles was determined using the headspace method and gas chromatographic analysis (Magen et al., 2014). Based on the obtained values,

conversion factors (ppm to nmol L$^{-1}$) were determined for each setup. For quality control, the regional boundaries were set to 1–500 nmol L$^{-1}$ (see section data management).

Atmospheric $CH_4$ was measured on board of the research vessels: for Stern-3 on the Littorina, Ludwig Prandtl and Uthörn with a Picarro G2301, a Picarro G2301, and a Microportable Greenhouse Gas Analyzer (LosGatos) respectively; for Stern-5 with a Picarro G2301, a Licor LI-8100A and a LosGatos GGA-911, respectively. All

data were corrected by the instruments for water vapor resulting in $CH_4$ dry values. The inlets for the instruments were approximately 4 m above the water surface and located either at the ship's bow (Littorina) or on a railing on the bridge (other ships). For quality control, the regional boundaries were set to 1.8–2.3 ppm (see section data management). Additional data for atmospheric $CH_4$ concentration were obtained from the meteorological station in Mace Head, Ireland, (https://gml.noaa.gov/dv/data/index.php?site=MHD), using the

monthly means of September 2019 and September 2020 (1.942 ppm and 1.957 ppm, respectively).



### 2.4    Calculation of the diffusive methane flux

The overall gas exchange across an air–water interface was determined according to (Wanninkhof et al., 2009) as:

$$F = k_{CH4} \cdot (c_m - c_{equ}) \tag{2}$$

where F is the rate of gas flux per unit area (mmol $m^{-2}$ $d^{-1}$), $c_m$ is the measured $CH_4$ concentration of the surface waters, and $c_{equ}$ is the atmospheric gas equilibrium concentration (Wiesenburg and Guinasso, 1979). For the atmospheric $CH_4$ concentration data, we either used our measured data or the data from the Mace Head observatory.

The gas exchange coefficient (k) is a function of water surface agitation. The k value in oceans and estuaries is determined mostly by wind speed ($U_{10}$). The determination of k is crucial to calculate the sea–air flux. We calculated the gas exchange velocity $k_{600}$ according to the following equation for coastal seas (Nightingale et al., 2000):

$$k_{600} = 0.333U_{10} + 0.222U_{10}^2 \tag{3}$$

We applied the wind-speed-based $k_{600}$ parameterization from Nightingale et al. (2000) here largely because it is commonly used and represents a compromise between relationships that have a very strong or a very weak wind-speed dependence (Yang et al., 2019).

The calculated $k_{600}$ (for $CO_2$ at 20°C) was converted to $k_{CH4}$ (Striegl et al., 2012), and the Schmidt number (Sc) was adjusted based on water temperature and salinity (Wanninkhof, 2014):

$$k_{CH4} / k_{600} = (Sc_{CH4} / Sc_{CO2})^{-0.5} \tag{4}$$

To determine the influence of wind and atmospheric $CH_4$ on the flux calculation, three combinations of data sets were applied Table 1):

-    Flux-1 with in-situ wind and in-situ atmospheric $CH_4$ concentrations, with a resolution of 1 minute.
-    Flux-2 with in-situ wind but with the atmospheric $CH_4$ concentrations from the station Mace Head, Ireland, with a resolution of 1 month.
-    Flux-3 fixed monthly atmospheric concentration from the station Mace Head and using hourly wind data from the German Meteorological Service.

Table 1: Calculation of the diffusive $CH_4$ flux with several combinations of data sources.

|  | Flux 1 | Flux 2 | Flux 3 |
|---|---|---|---|
| Dissolved $CH_4$ | In situ* | In situ* | In situ* |
| Atmospheric $CH_4$ | In situ* | Mace Head*** | Mace Head*** |
| Wind speed | In situ* | In situ* | DWD** |



temporal resolution of every minute*, every hour**, every month***

To improve the estimation of the diffusive $CH_4$ flux for the whole study area, we calculated an area weighted diffusive flux. We split the diffusive flux data (n = 941 for 2019 and 3028 for 2020) into groups with a bin size of 100 $\mu mol\ m^{-2}\ d^{-1}$ and calculated a frequency distribution of the mean diffusive flux classes (0-100, 100-200, 200-300 ….. $\mu mol\ m^{-2}\ d^{-1}$). Next, the relative area was calculated by multiplying the relative frequency of each class with the total area. Then, the relative area of each class was multiplied with the respective diffusive flux to

obtain the relative areal flux. The sum of all relative areal fluxes finally resulted in the total weighted flux of the whole area. The standard deviation was determined from the relative areal fluxes. An example of the calculation is given in the supplementary Table S1.

To enhance the validity of our results, we extrapolated our calculated diffusive fluxes from our respective study

areas to areas in accordance with the ecosystem type classification of the German Federal Statistical Office (Statistisches Bundesamt ((Destatis), 2021), which assigns all areas of Germany to different ecosystem types without gaps or overlaps (https://oekosystematlas-ugr.destatis.de/). We used the following ecosystems, overlapping with our cruise track: eastern Wadden Sea of the Weser River (490000003), open coastal sea of the Weser (490000004), coastal sea of the Weser (490000005), Helgoland (590000002), coastal sea of the Elbe

River (590000003), western Wadden Sea of the Elbe (590000005), Outer Elbe North (590000006) and Piep Tidal basin (950000001, Fig. S1). These ecosystems cover a total area of $3.78 \times 10^9\ m^2$ (377947 ha).

### 2.5    Data management and handling

During the cruises or shortly afterwards, all data from all ships were uploaded to AWI's data web service

(Koppe et al., 2015), https://dashboard.awi.de/data-ingest/index.html#) at the highest available resolution. From this repository, data from different sensors can be combined, aggregated over time and downloaded as .csv files. In a second step, we applied a quality and plausibility control procedure to the data. In a first plausibility procedure, the ARGO algorithms (Bittig et al., 2019) for data quality flagging (manufacturer range, local range, spike check and gradient check) were applied assigning a bad data flag to values outside the ranges.

Additionally, as previous cruises had shown that it is essential to compare and possibly correct the sensor's data between the vessels (Bussmann et al., 2021b; Fischer et al., 2021), two, respectively four, intercalibration phases were scheduled during Stern-3 and Stern-5. During these phases, all vessels were in close proximity to each other (100–600 m) with all underway systems running and sampling the same water body.

In a machine-learning supported expert analysis (Fischer et al., 2021), sensor data of all three ships were first

visualized synoptically for the full time of the cruise and for the intercalibration intervals. From these comparisons, correction factors for those sensor data with significant accuracy deviations during the intercalibration phases were calculated and applied to the ships' respective sensor data. For example, on Stern-5, the water temperature from Littorina was used as a lead sensor as confirmed by precise measurements. As the temperature data from RV Ludwig Prandtl deviated by -0.03° during the intercalibration phases, +0.03°C were

added to those temperature data. In contrast, the temperature data from RV Mya deviated for about + 0.07° from the reference value (from Littorina), therefore -0.07° were subtracted from RV Mya's temperature data. For subsequent calculations, all data were used with one minute resolution. All calculations and statistics were



performed with R-studio (version 2023.09.01+494, Posit Software, PBC). The combined and corrected datasets, including the details of correction, can be found at the online repository pangea.de

(https://doi.pangaea.de/10.1594/PANGAEA.962691 for Stern-5, and ## for Stern-3, submitted).

## 3  Results

### 3.1  Oceanographic and meteorological conditions in September 2019 (Stern-3)

Water temperatures in the study area in September 2019 ranged from 17.1 to 19.7°C, salinity ranged from 18.6 to 33.1, and oxygen saturation ranged from 80.1 to 100.4%. The meteorological situation differed substantially between 10 and 11 September, with a mean wind speed of 6.7 m s$^{-1}$ compared to 9.2 m s$^{-1}$, and the wind direction shifted from west and west-northwest to southwest, respectively (Fig. 3).

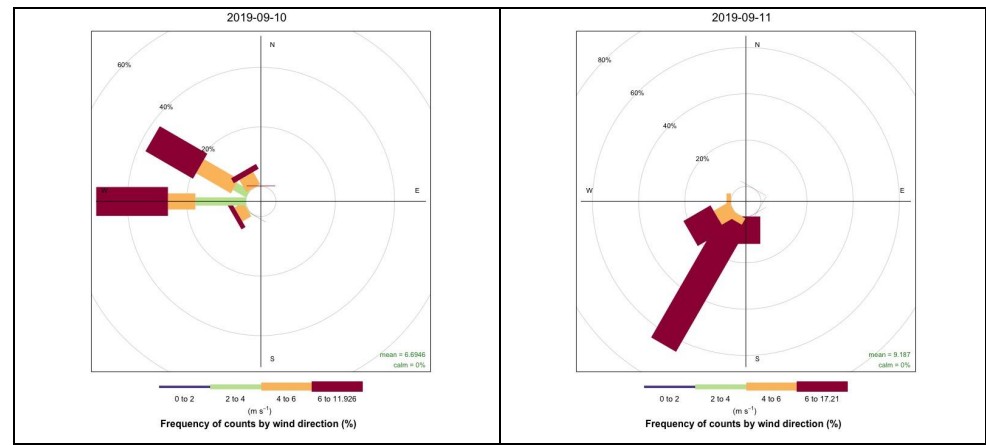

**Fig. 3 Wind rose with wind speed and direction for 10 September (left) and 11 September 2019 (right, Stern-3).**

As the diffusive flux was calculated using the wind speed data, the CH$_4$ related parameters are also described separately for the two days. On 10 September, dissolved CH$_4$ concentrations showed a median of 22.6 nmol L$^{-1}$ (range 3.9–304.9 nmol L$^{-1}$). High CH$_4$ concentrations were encountered near Cuxhaven, and at two to three

patches between Helgoland and Cuxhaven (Fig. 4). On 11 September, concentrations were lower, with a median of 12.3 nmol L$^{-1}$ (range 1.1–175.8 nmol L$^{-1}$). Lowest concentrations (1–2 nmol L$^{-1}$) were encountered west of the islands of Scharhörn and Neuwerk.

Atmospheric CH$_4$ concentrations had a median of 1.949 ppm (range 1.936 to 1.971ppm) on 10 September versus a median of 2.064 ppm on 11 September (range 1.948–2.255 ppm). On 11 September, rather high values

(2.15 ppm) were observed near the island of Scharhörn. As the atmospheric CO$_2$ data were not elevated at this site, the influence of ship exhausts can be excluded. The wind was coming from south-southwest (200°), and the tide was just increasing. We assume that the air mass crossing our cruise track had an inherent natural variability.

The diffusive flux was first calculated with in-situ wind and in-situ atmospheric CH$_4$ concentrations (flux-1).

For both days combined, the average flux was 221 ± 351 µmol m$^{-2}$ d$^{-1}$, the median was 97 µmol m$^{-2}$ d$^{-1}$, and the range was from -27 to 2342 µmol m$^{-2}$ d$^{-1}$. On 10 September, the diffusive flux had a median of 131 µmol m$^{-2}$ d$^{-1}$



(range 316–1500 µmol m$^{-2}$ d$^{-1}$), with lowest values near Helgoland and near the island of Scharhörn (Fig. 4, left). Highest values were observed southeast of Helgoland. On 11 September, the diffusive flux was half of the one on the day before with a median of 62 µmol m$^{-2}$ d$^{-1}$ (range -27 – 2342 µmol m$^{-2}$ d$^{-1}$). Highest values were

observed again in the region between Helgoland and Cuxhaven; lowest and negative values were observed west of Cuxhaven (Fig. 4 right).

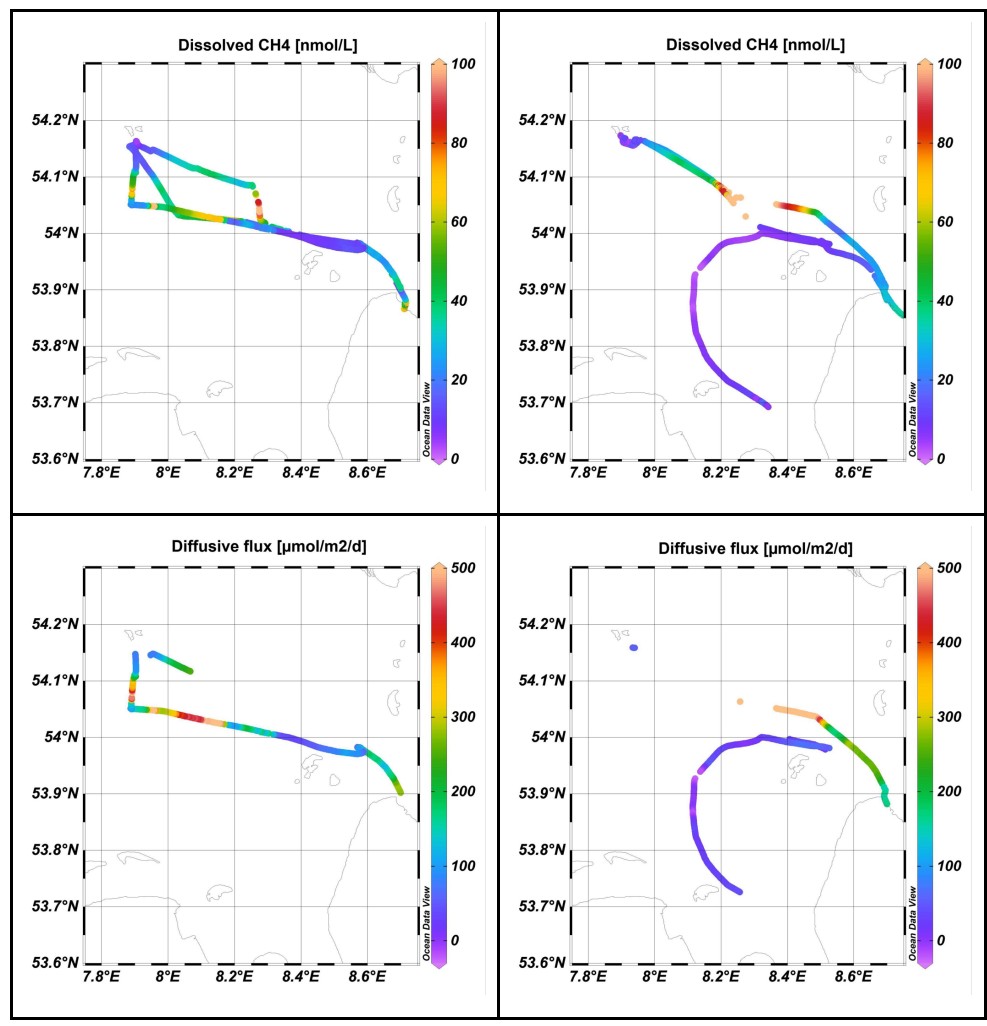





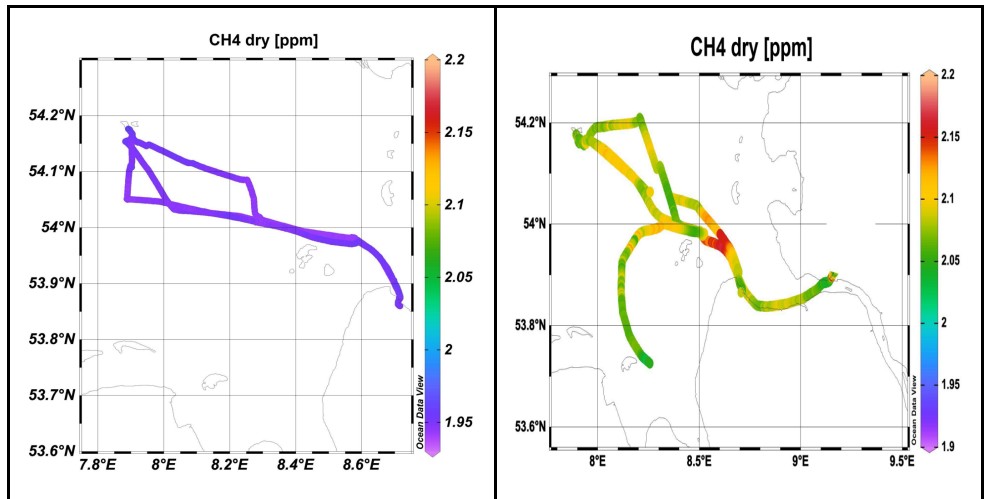

**Fig. 4: Concentrations of dissolved CH₄ (top), diffusive CH₄ flux (middle) and in-situ atmospheric CH₄ concentrations (bottom) on 10 September (left) and 11 September (right).**


### 3.2     Oceanographic and meteorological conditions in September 2020 (Stern-5)

Water temperatures in the study area in September 2020 were warmer compared to 2019, ranging from 17.6 to 21.4°C, salinity ranged from 13.8 to 33.4 and oxygen saturation ranged from 70 to 109%.

The meteorological situation differed substantially between the sampling dates, and therefore the data are

presented per day (and not for the whole area). On 31 August, the median wind speed was 1.9 m s⁻¹ coming from the north-northwest. On 1 September, the wind direction shifted towards northeast, with a median speed of 4.5 m s⁻¹. On 2 September, wind speed decreased to 2.2 m s⁻¹ with no preferred direction. On 3 September, the wind freshened to a median of 8.2 m s⁻¹ and was blowing from the south and south-southwest (Fig. 5).

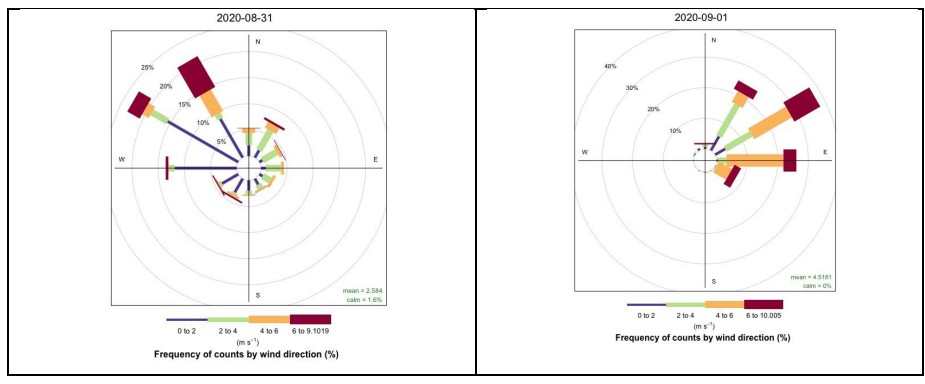





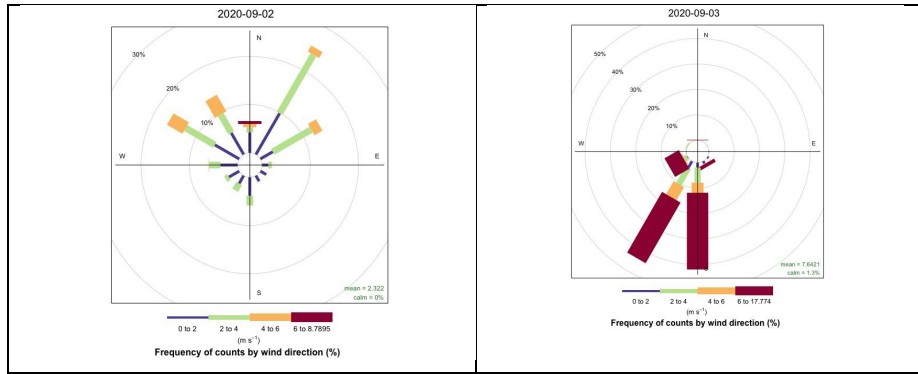

**Figure 5: Wind rose with wind speed and direction for 31 August to 3 September 2020 (Stern-5).**

Dissolved $CH_4$ concentrations for all days ranged from 1.4 to 607.9 nmol $L^{-1}$, with a median of 26.2 nmol $L^{-1}$ (Fig. 6 left). Low concentrations of dissolved $CH_4$ (5–10 nmol $L^{-1}$) were observed southwest of Helgoland, in the outer Weser Estuary and in the northern region of the study area towards the island of Sylt (Fig. 6a). West of Büsum, we observed an area of high concentrations, as well as near Amrum, i.e., near the North Frisian Wadden Sea (> 100 nmol $L^{-1}$), with additional patches of elevated concentrations (70–100 nmol $L^{-1}$) located, for example, east and north of Helgoland.

The average diffusive $CH_4$ flux was 159 ± 444 μmol $m^{-2}$ $d^{-1}$. The median diffusive $CH_4$ flux for all days combined was 61 μmol $m^{-2}$ $d^{-1}$, ranging from 0.2–4645 μmol $m^{-2}$ $d^{-1}$. The spatial distribution of the flux was mostly a mirror image of the dissolved $CH_4$ concentration (Fig. 6 right). For the dataset from RV Littorina no flux data were calculated, as no wind data were available. The data for dissolved and atmospheric $CH_4$ and the diffusive $CH_4$ flux for the individual days are shown in Figure S2, analogous to Fig. 4.





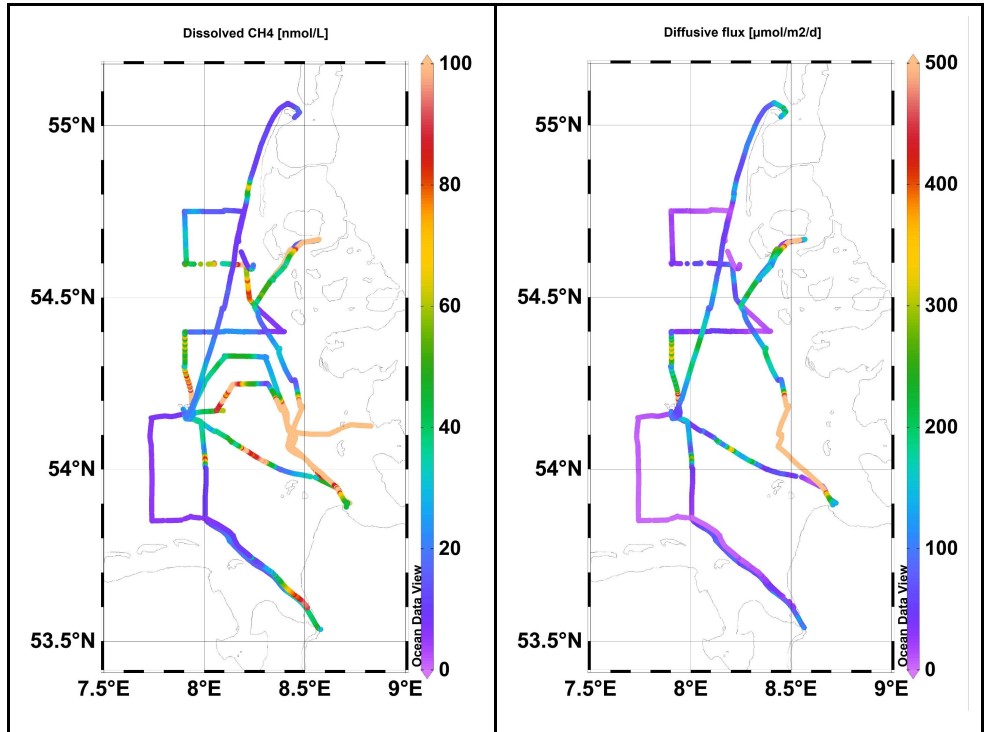

**Figure 6: Concentrations of dissolved CH₄ (left) and the diffusive CH₄ flux (right) for the whole study area and**

**Stern-5 study period (30 August to 3 September 2020).**





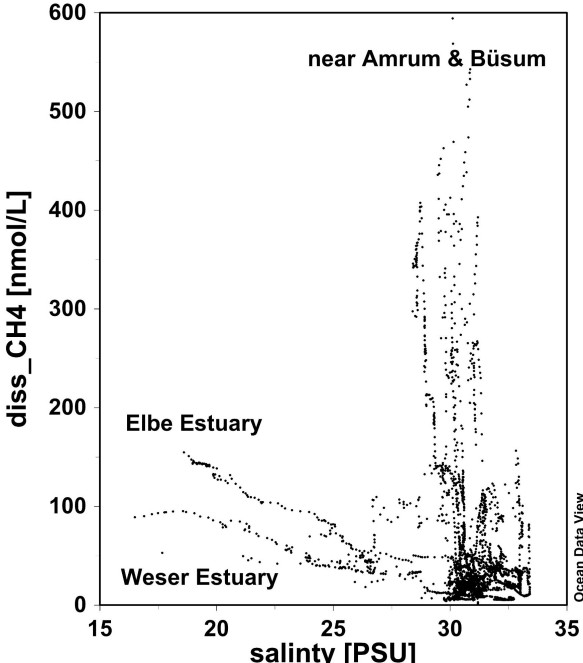

**Fig. 7: Dissolved** $CH_4$ **concentration plotted versus salinity for September 2020 (Stern-5). The geographic location of**
**the data pairs is also indicated.**

In both the Weser and Elbe estuaries, methane-rich river water was diluted with methane-poor marine waters
(Fig. 7). The riverine endmember of the Weser showed lower $CH_4$ concentrations (95 nmol L$^{-1}$) than the Elbe
endmember (151 nmol L$^{-1}$). However, highest $CH_4$ concentrations coincided with high salinities (> 30). These
concentrations were all observed in the area west of Amrum and Büsum in the Wadden Sea. Thus, these areas
were clearly not part of the dilution scheme, but a strong source of $CH_4$.

The median atmospheric $CH_4$ concentrations increased during the observed time span, from 1.951 ppm on 31
August, 1.979 ppm on 1 September, 2.022 ppm on 2 September to 2.078 ppm on 3 September (Fig 8). This
increase over time was especially evident when comparing 1 and 2 September. The same area was covered (as
the ships were returning to the same ports), and a substantial increase was observed, especially near the coast.





**Figure 8: Atmospheric CH$_4$ concentration for 30 August to 3 September 2020 (Stern-5). The black boxes mark the areas where the vessels approached or left the Wadden Sea area.**

### 3.3    Calculations of diffusive fluxes using in-situ and land station-based data

The calculation of the diffusive CH$_4$ flux was performed with three combinations of datasets to explore the influence that the use of atmospheric background data (for CH$_4$) and the closest land stations (for wind) has on CH$_4$ flux results (Table 1). For a better assignment of the wind data, the area was split into three boxes, one near Helgoland, one near Bremerhaven and one near Cuxhaven (see Fig. 1 and 2).

For the September 2019 cruises, the median in-situ atmospheric CH$_4$ concentrations ranged from 1.950 to 2.060 ppm, compared to a monthly mean of 1.942 ppm at the Station Mace Head (Table 2). For the wind speed, there was no or only a small difference between the in-situ data and data from the weather stations in Bremerhaven and Helgoland, while for the station Cuxhaven wind speed was almost 4 m s$^{-1}$ lower than the in-situ wind speed. Higher atmospheric CH$_4$ concentrations lead to higher equilibrium concentrations of dissolved CH$_4$, and therewith to a smaller oversaturation and lower diffusive fluxes. In 2019, the measured atmospheric CH$_4$ concentrations were always higher than those at the meteorological station. Consequently, the flux-2 values were often slightly higher than the flux 1 values (Fig. S3). The strongest difference was noticeable when comparing flux-1 with flux-3. When station data were used for both atmospheric CH$_4$ and wind (flux-3), there were substantial differences from the calculations using only in-situ data (flux-1).

The median in-situ atmospheric CH$_4$, concentrations ranged from 1.967 to 1.994 ppm for the September 2020 cruises, encompassing the monthly mean of 1.987 ppm at the Station Mace Head (Table 2). The wind speed measured on board the vessels was always higher than the data from the stations, with a difference of 0.4 and 0.5 m s$^{-1}$ for Bremerhaven and Cuxhaven, and a difference of 0.8 m s$^{-1}$ for Helgoland, resulting in comparatively lower flux-3 data. The flux-1 and flux-2 data were similar or almost identical, while flux-3 data were clearly different or lower than the other two fluxes (Fig. 9). For both years, the number of flux-3 data was higher than for flux-1 and flux-2. The wind data for flux-3 were taken from the meteorological station with hourly averages, while the in-situ wind data were measured every minute, but with data gaps due to the quality control of the data.

**Table 2: The median concentrations of dissolved and atmospheric CH$_4$ and median wind speed in September 2019 and September 2020, calculated either with in-situ data, with monthly mean data from Station Mace Head for atmospheric CH$_4$ or as hourly mean wind data from three weather stations at Bremerhaven, Cuxhaven and Helgoland. The calculation of the diffusive flux was performed according to Table 1. The flux calculations were performed for an area near Bremerhaven (area B), an area near Cuxhaven (area C) and an area near Helgoland (area H), see also Figures 1 and 2.**

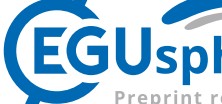

| | Dissolved CH$_4$ in situ | Atmospheric CH$_4$ in situ / Mace Head | Wind speed in situ / meteo. station | Flux-1 | Flux-2 | Flux-3 |
|---|---|---|---|---|---|---|
| | nmol L$^{-1}$ | ppm | m s$^{-1}$ | μmol m$^{-2}$ d$^{-1}$ | | |
| 2019 | median | median | median | mean ± SD (n) | | |
| Area B | 8.0 | 2.060 / 1.942 | 9.5 / 9.5 | 32 ± 4; (51) | 36 ± 8; (64) | 38 ± 15; (65) |
| Area C | 14.2 | 1.950 / 1.942 | 10.4 / 6.5 | 138 ± 78; (227) | 142 ± 78; (293) | 50 ± 25; (493) |
| Area H | 24.1 | 1.956 / 1.942 | 7.5 / 7.5 | 184 ± 192; (281) | 185 ± 190; (281) | 105 ± 88; (514) |
| 2020 | | | | | | |
| Area B | 22.6 | 1.994 / 1.957 | 3.2 / 2.7 | 52 ± 36; (269) | 52 ± 36; (282) | 23 ± 17; (269) |
| Area C | 79.6 | 1.967 / 1.957 | 2.7 / 2.3 | 235 ± 342; (226) | 235 ± 342; (226) | 57 ± 38; (244) |
| Area H | 26.4 | 1.991 / 1.957 | 5.9 / 5.1 | 88 ± 92; (827) | 90 ± 92; (851) | 57 ± 62; (1186) |


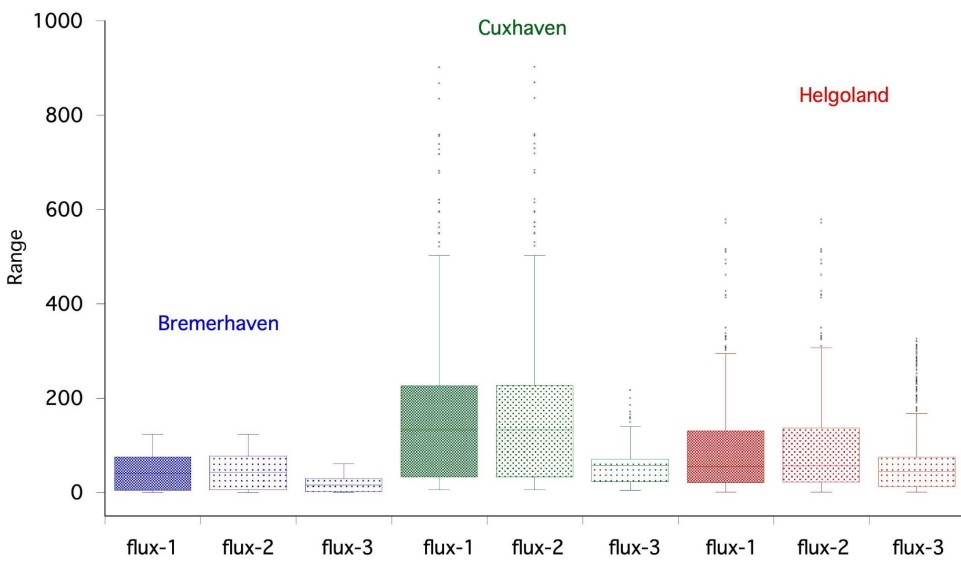

**Figure 9: Range of diffusive CH$_4$ fluxes calculated with all in-situ data from the cruises in September 2020 (flux-1),**
**with in-situ data and atmospheric CH$_4$ data from the land station (flux-2), and with in-situ data and atmospheric CH$_4$**
**concentration and wind from three land stations (flux-3). The calculations were performed for the region of**
**Bremerhaven (blue), for Cuxhaven (green) and Helgoland (red, see Fig. 2)**

### 3.4 Area-weighted calculation of the diffusive methane flux

The frequency distribution for the 2019 flux data is shown in Fig. S4. The majority of flux data (48%) belonged
to the range class 0–100 μmol m$^{-2}$ d$^{-1}$. The subsequent classes (100–500 μmol m$^{-2}$ d$^{-1}$) had frequencies between



19.2% and 3.6%. Fluxes higher than 500 µmol m$^{-2}$ d$^{-1}$ had a total frequency of 9%. Negative fluxes (-100–0 µmol m$^{-2}$ d$^{-1}$) occurred with a frequency of 3%.

The frequency distribution for the 2020 flux data is shown in Fig. S5. Again, most of the data were found in the
class 0–100 µmol m$^{-2}$ d$^{-1}$, with a frequency of 67%. The other classes (100–500 µmol m$^{-2}$ d$^{-1}$) showed frequencies between 21.0% and 0.6%. Fluxes higher than 500 µmol m$^{-2}$ d$^{-1}$ had a total frequency of 5%. In contrast to the September 2019 values, negative fluxes (in the range of -100–0 µmol m$^{-2}$ d$^{-1}$) were not observed. To calculate the weighted flux for our study area, we related the total area of the Helgoland Bay (3.78 x 10$^9$ m$^2$) to the frequency of the flux classes, as explained in the Method section and in supplementary Table S1. This
resulted in the total area-weighted diffusive fluxes of 836 ± 32 and 600 ± 21 kmol d$^{-1}$ for the area of Helgoland Bay for 2019 and 2020, respectively.

The other approach was to multiply the median or mean of all flux data with the total area. The standard deviation for the mean was also multiplied with the area to maintain the same unit. This resulted in much lower values for the median flux data and identical values for the mean flux data, compared to the area-weighted
approach. However, the standard deviation of the mean flux data was much larger than for the area-weighted approach (Table 3).

Table 3. Comparison of three approaches to calculate the total diffusive flux from Helgoland Bay with an area of 3.78 x 10$^9$ m$^2$.

|  | Area related diffusive flux in Helgoland Bay (kmol/d) | | |
|---|---|---|---|
|  | median (range) | mean ± SD | Area-weighted |
| Sept 2019 | 365 (-104–8851) | 836 ± 1328 | 836 ± 32 |
| Sept 2020 | 229 (1–17558) | 600 ± 1676 | 600 ± 21 |


### 3.5    Estimation of flux contributions to atmospheric concentrations

We often observed a substantial increase in atmospheric CH$_4$ concentration between single days, but the source of the additional atmospheric CH$_4$ was unclear. Sources could be the diffusive flux from the sea into the atmosphere or different origins of the air masses above the water.
In September 2019, we observed an increase in atmospheric CH$_4$ concentrations of 0.116 ppm between 10 and 11 September (from 1.950 to 2.065 ppm). The average diffusive flux on these two days was 222 µmol m$^{-2}$ d$^{-1}$. Using the ideal gas law, we converted the CH$_4$ flux into a gas volume (air temperature of 16°C, pressure of 1015 mbar, time frame one day, comparable to the calculations of (Zang et al., 2020). Under the idealized assumption that we had no advective exchange, the diffusive flux alone would explain the observed concentration difference
for a mixed atmospheric layer of 45 m in height.

In September 2020, the strongest difference in atmospheric CH$_4$ was observed between 1 and 2 September (delta = 0.079 ppm). The average diffusive flux for these two days was 48 µmol m$^{-2}$ d$^{-1}$. The calculation for the



idealized mixed layer height for this day yielded 15 m. Further assuming a planetary boundary layer height for mid-latitudes of about 300 m (a lower range estimate based on climatologies by (Ao et al., 2012) and a well-
mixed surface layer of about 10% thereof, i.e., 30 m, the observed increase of atmospheric $CH_4$ in September 2019 could have been mainly due to diffusive flux from the sea into the atmosphere. In September 2020, however, the calculated height of the mixed surface layer would be 15 m, which is not realistic. Thus, the observed increase of atmospheric $CH_4$ has to be attributed mainly to advection.

A linear regression analysis between the diffusive flux and atmospheric $CH_4$ revealed no significant correlation;
also, when split by single dates. Strong wind results in an increased diffusive flux, but as the mixing of the atmospheric $CH_4$ also increases, the signal of the diffusive $CH_4$ imported will be "diluted". Thus, we tested the hypothesis that only under low wind conditions a correlation would be detectable. Therefore, we split the wind in classes (<10, <9, <8 m s$^{-1}$ etc.) and repeated the analyses for each class. These class-separated calculations revealed no correlation between the diffusive flux and atmospheric $CH_4$ concentration at wind speeds >5 m s$^{-1}$.
However, at wind speeds less than 5 m s$^{-1}$, a significant correlation between diffusive flux and atmospheric $CH_4$ was detected ($r^2 = 0.52$). The strongest correlation was detected at wind speeds <2 m s$^{-1}$ ($r^2 = 0.75$).

A further possible cause for increases in atmospheric $CH_4$ concentrations are changes in advection under the assumption that wind coming across the sea has a lower $CH_4$ content than wind coming from land. As wind direction is no linear parameter, we divided the parameter into 30° classes, followed by a one-way analysis of
variance. It revealed that the wind direction had a significant influence on the atmospheric $CH_4$ concentrations in our two study periods of September 2019 and September 2020 (p < 0.001).

In September 2019, the highest atmospheric $CH_4$ concentrations were observed when the wind came from the south-southwest (210–240°) with a median of 2.08 ppm, and lowest values were observed when the wind came from northerly directions (0–30°, 300–330°, 330–360°) with a median of 1.94 ppm, 1.95 ppm and 1.95 ppm,
respectively. In September 2020, highest atmospheric $CH_4$ concentrations were observed when the wind came from the south (180–210°) with a median of 2.07 ppm, and lowest values were observed when the wind came from the east (90–120°) with a median of 1.98 ppm.

In addition to the wind signal, we looked for a possible tidal impact. On 1 September 2020, the RV Ludwig Prandtl approached the harbor at Wyk auf Föhr from 14:00–16:00 UTC and left port around 04:00–06:00 UTC
the following morning (Fig. 8). The wind was blowing from northeast on both occasions, with rather low wind speed of less than 5 m s$^{-1}$ (Fig. 5). High concentrations of dissolved $CH_4$ were observed, during the approach of the harbor (the data points outside the dilution scheme of Fig. 7). The diffusive $CH_4$ flux increased, and a most pronounced increase of 0.186 ppm of atmospheric $CH_4$ was observed (Table 4). In contrast to the overall analyses described above, for this areal section neither wind speed nor diffusive flux were correlated with
atmospheric $CH_4$. However, as Wyk auf Föhr is surrounded by the tidal flats of the Wadden Sea, at low tide these flats are exposed to the atmosphere and tidal creeks withdraw pore water from the surroundings, which results in increased atmospheric $CH_4$ due to the release of $CH_4$ formed through anaerobic processes. A similar pattern of atmospheric $CH_4$ was observed for the tidal area off Büsum in the cruise section covered by RV Littorina. Atmospheric $CH_4$ increased for this section from 1.975 to 2.193 ppm, however no additional data
(wind, diffusive $CH_4$ flux) are available for this ship.

**Table 4: Atmospheric $CH_4$ concentration in relation to tidal state, wind and diffusive $CH_4$ flux.**




| Section toward Wyk/Föhr | 1 September 2020 14:00–16:00 UTC | 2 September 2020 04:00–06:00 UTC |
|---|---|---|
| Atmospheric $CH_4$ (ppm) | 1.971 ±0.007 | 2.157 ± 0.04 |
| Diffusive flux ($\mu mol\ m^{-2}\ d^{-1}$) | 24 ± 15.9 | 115 ± 22 |
| Wind speed ($m\ s^{-1}$) | 4.6 ± 0.5 | 3.1 ± 0.4 |
| Wind direction (°) | 63 ± 4 | 44 ±12 |
| Tide at Wyk | HT at 12:39 | LT at 07:28 |
|  |  |  |
| Section towards Büsum | 1 September 2020 10:40–11:40 UTC | 2 September 2020 04:40–05:40 UTC |
| Atmospheric $CH_4$ (ppm) | 1.982 ±0.0072 | 2.140 ± 0.02 |
| Tide at Büsum | HT at 11:24 | LT at 05:53 |

**4    Discussion**

**4.1    Error discussion for calculations of diffusive fluxes using different source data**

In this study we applied several different methods to calculate diffusive sea-to-air $CH_4$ fluxes, either by using different databases for local values or by applying a weighted method for $CH_4$ fluxes for larger areas.

The comparison between results obtained using different data sources showed that the choice between using
atmospheric $CH_4$ concentrations from in-situ data or from a land station had no large effect (flux-1 versus flux-2, Fig. 9). In our study, the monthly average atmospheric $CH_4$ concentration from Mace Head station, that is usually used for providing atmospheric background concentrations, always showed lower values than our in-situ data. Thus, the saturation concentration of dissolved $CH_4$ was lower, resulting in a smaller sea-to-air diffusive flux. However, in relation to the variability of the measured datasets, this difference was minor. The calculated
flux-2 values reached on average 103 ± 6% of the corresponding flux-1 values (n = 6).

In contrast, it was important to use the in-situ wind speed for calculating the $CH_4$ fluxes (flux-3) instead of using data from the closest meteorological land stations. In some cases, the wind data were nearly identical, in other cases the wind data from the land monitoring stations were lower, resulting in significantly smaller diffusive fluxes. The stronger impact of the wind speed is based on the fact that the flux calculation uses a quadratic wind
speed formulation (equation 2, (Nightingale et al., 2000). Relating the flux-3 (land station) data to the flux-1 (in-situ) data revealed that the flux-3 values only reached an average of 58% ± 34% of the flux-1 values (n = 6). As our flux data have a high variability (see below), a high temporal resolution of the data (as in flux-1) is favorable over hourly (wind data) or monthly (atmospheric $CH_4$ data) resolution. Several combinations of in-situ data and data from databases have been reported in literature for calculating diffusive sea-to-air $CH_4$ fluxes (Myllykangas
et al., 2020; Woszczyk and Schubert, 2021; Mau et al., 2015; Bussmann et al., 2021b; Humborg et al., 2019). Based on our direct comparison of these different approaches, we strongly recommend obtaining in-situ wind data.

We furthermore observed a high variability in all diffusive $CH_4$ flux data. For the entire 2019 dataset, the average diffuse $CH_4$ flux was 221 ± 351 $\mu mol\ m^{-2}\ d^{-1}$ (n = 941), and for 2020 it was 159 ± 444 $\mu mol\ m^{-2}\ d^{-1}$ (n =
3028). The coefficient of variation (CV) was 158% and 279%, respectively. Flux values for $CH_4$ in general have



a high variability (36-71% (de Groot et al., 2023); 73% (Bussmann et al., 2021b), 78% (Humborg et al., 2019)), so that the CV values found here are not unusual. However, to avoid a possible elevated flux variability due to a low sample size (methodological error), we applied a modified bootstrap analysis on the 2020 data to elucidate the effect of sample size on the calculated flux variability. The 2020 dataset had a total of 3028 measurements.

From this dataset, we iteratively drew random subsamples, beginning with 20 values and increasing the sample size by 10 and 100 in each iteration. By this method, we finally received 39 datasets with an increasing number of flux values, starting at 20 and reaching up to 3028 values. The mean of these 39 datasets was calculated and plotted versus sample size (Fig. 10). The analysis revealed that the calculated average mean flux was independent of the sample size (slope, t= 0.02, p = 0.98) and that the variability of the flux values remained

stable between 104 and 190, around an average mean value of 156 for a sample size of about 900-1000 or higher. This supported our presumption that our sample size of 3028 for the year 2020 was sufficiently high to avoid a sample size bias in flux variability and represented a realistic system flux in the area. This high spatial variability is also evident in Figure 6. Thus, it is debatable if our study area is a uniform area and if it is reasonable to average the diffusive flux for the whole study area.


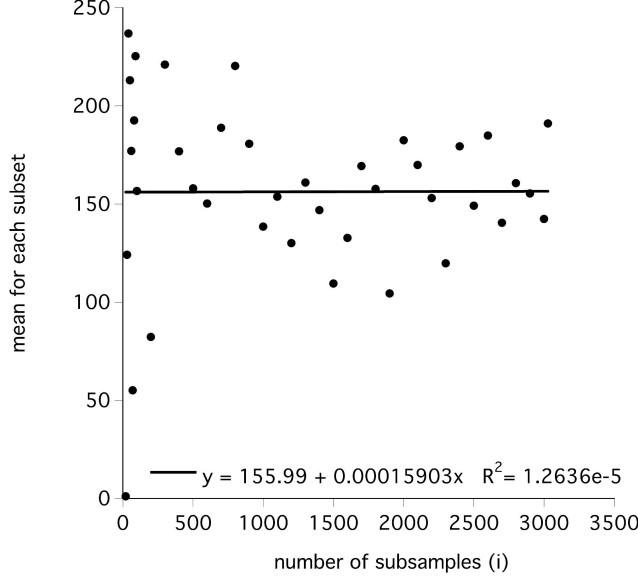

**Figure 10: The mean of the diffusive CH₄ flux calculated for a random number of *i* subsamples. The black line**
**indicates the regression line.**

**4.2     Area-weighted calculation of the diffusive CH₄ flux**

To extrapolate the diffusive flux for larger areas (in mol d⁻¹), the general approach is to multiply the target area
(m²) with the median or mean flux (mol d⁻¹ m⁻²) calculated from a restricted number of samples in the area. Figures S4 and S5 show the data base and frequency distribution for such a calculation for our data from the



years 2019 and 2020. Both figures show a highly skewed distribution of flux values, with values <100 µmol m$^{-2}$ d$^{-1}$ having the largest share resulting in a skewness of 3.4 and 6.1 for 2019 and 2020, respectively. Table 3 shows the results of different methods of averaging when calculating the diffusive flux for our target areas in

2019 and 2020.

According to common recommendations for data with a positive skew (data with the frequency distribution shifted to the left side), the median is always smaller than the mean (median < mean) (Köhler et al., 1996; Doane and Seward, 2011). Accordingly, the median in our calculations revealed almost 3-fold lower overall flux estimates compared to the mean values. To circumvent this bias, a variety of terrestrial, marine and limnic

studies (Mallast et al., 2020; Li et al., 2020; Baliña et al., 2023) stressed the importance of applying an area-weighted approach for up-scaling $CO_2$ flux data. We therefore also applied this method to our data (Table 3, right column). This calculation revealed that flux values for the area calculated by using the arithmetic mean are identical to the area-weighted mean values and significantly higher than the median-based average flux values. In addition, the SD from the area-weighted flux was much lower. Thus, if area-weighted flux estimations are not

possible due to a limited dataset, our data suggest that using the mean value is the preferred procedure rather than the median-based average flux calculations for an area.

### 4.3    Methane emissions from Helgoland Bay

In our study we revealed mean diffusive fluxes of 221± 351 µmol m$^{-2}$ d$^{-1}$ and 159 ± 444 µmol m$^{-2}$ d$^{-1}$ and

median values of 97 and 61 µmol m$^{-2}$ d$^{-1}$ for 2019 and 2020, respectively

These numbers are within the same range as published previously for June 2019 (65 µmol m$^{-2}$ d$^{-1}$) (Bussmann et al., 2021b). However, higher flux data are reported for autumn and winter, 104 µmol m$^{-2}$ d$^{-1}$ for the Dogger bank area (Mau et al., 2015) and 124–299 µmol m$^{-2}$ d$^{-1}$ for our study area (Winkler, 2019) (all median values). These higher fluxes were explained by the authors with higher wind velocities in the autumn / winter season. In

comparison to other coastal seas our diffusive fluxes are similar to fluxes from the Belgian North Sea (161 – 221 µmol m$^{-2}$ d$^{-1}$ (Borges et al., 2019). In contrast much higher fluxes are reported from the Baltic Sea, with -9 – 3110 µmol m$^{-2}$ d$^{-1}$ by Gutiérrez-Loza et al. (2019) and 2400 µmol m$^{-2}$ d$^{-1}$ by Humborg et al. (2019). On the other side, low fluxes are reported from the Atlantic coast of Spain (7 – 20 µmol m$^{-2}$ d$^{-1}$; Ortega et al., 2023) and costal Chile (5 ± 5 µmol m$^{-2}$ d$^{-1}$; Farías et al., 2021)).

However, these comparisons are all based on the median values for the whole area. When focusing on specific locations, other patterns become evident. The lowest fluxes, i.e., negative fluxes, were observed in September 2019 in an area west of Cuxhaven (-15 to -27 µmol m$^{-2}$ d$^{-1}$, Fig. 4b). At this site and at this time, the water was shallow (< 5 m) with strong winds from southwest resulting in short waves. Thus, we assume that this water body was mostly depleted of $CH_4$ and was acting as $CH_4$ sink. For the open North Sea, other studies have

reported undersaturation of dissolved $CH_4$ (Upstill-Goddard et al., 2000) (Bange, 2006), but not for nearshore areas like in this study.

Highest $CH_4$ concentrations were observed in the Wadden Sea and not related to river water inputs (Fig. 7). Elevated diffusive fluxes and elevated atmospheric concentrations were also observed at these locations and especially at low tide (Tab. 3). Sand flats or tidal flats are known to be a source of biogenic $CH_4$ that is released

directly into the water column of the North Sea and to the atmosphere during low tide (Wu et al., 2015; Beck and Brumsack, 2012). It has been reported that peaks in $CH_4$ coincided with ebb tides at multiple sites located



along the flanks of the estuary adjacent to tidal flats and wetlands (Pfeiffer-Herbert et al., 2019; Trifunovic et al., 2020).

**4.4 Estimation of sea-air flux contribution to atmospheric concentrations**

The average atmospheric $CH_4$ concentrations in this study were $2.03 \pm 0.08$ ppm and $2.05 \pm 0.08$ ppm for September 2019 and 2020, respectively. This is about 0.06 ppm higher than the values from Mace Head at the west coast of Ireland. Since 2020, there have been two new ICOS-Stations on Helgoland and Sylt, which are located in or near our study area. Their average September data (2021 and 2022) support our elevated

atmospheric $CH_4$ concentrations with $2.04 \pm 0.07$ and $2.04 \pm 0.08$ ppm (Kubistin et al., 2023; Couret and Schmidt, 2023).

One aim of our study was to clarify if, or under which circumstances, the diffusive flux from the water is detectable in the atmosphere above. An increased wind speed will lead to an increased $CH_4$ flux into the atmosphere above the water, however, this process is counteracted by the fact that increasing wind speed also

leads to increased mixing of the atmosphere and any input will be quickly "diluted".

Our study has shown that there is only a significant correlation between $CH_4$ fluxes and atmospheric $CH_4$ concentrations at wind speeds < 5 m s$^{-1}$. For two examples, when we observed a strong increase of atmospheric $CH_4$, we estimated whether this increase was due to $CH_4$ input from the sea. For September 2019, the increase in atmospheric $CH_4$ could be attributed to the input from the sea, but not for September 2020. Thus, the static

approach of (Zang et al., 2020) could not be confirmed by our data, as this approach does not take the turbulent mixing of the atmosphere into account. Several other studies, which also measured atmospheric $CH_4$ concentrations and diffusive fluxes simultaneously, confirm this absent relationship (Myhre et al., 2016; de Groot et al., 2023; Gutiérrez-Loza et al., 2019). From our more detailed estimates, we conclude that the comparison of diffusive fluxes and atmospheric concentrations alone does not account for the interactions of

diffusive flux and atmospheric convection.

Wind direction or advection of air masses have a strong influence on atmospheric $CH_4$ concentrations (Pankratova et al., 2022; Yang et al., 2019). Our data show that, when the wind was coming from the south or south-southwest, significantly higher atmospheric $CH_4$ concentrations were observed (2.07–2.08 ppm). Wind from this direction originated from the German mainland with the ports of Bremerhaven and Wilhelmshaven,

and from regions with intensive livestock farming. Our air mass origin assumptions are supported by results from the NOAA back trajectories modeling (at 10 m height, https://www.ready.noaa.gov/hypub-bin/trajasrc.pl). Air masses at the end of 3 September 2020 originated from the mainland of Lower Saxony as well as the Netherlands (Fig. S6). On the other hand, lowest atmospheric $CH_4$ values were observed when the wind was blowing from the north (1.95 ppm) in 2019. Wind from this direction originates from the open North Sea and

shows similar values as those observed at the NOAA Mace Head station in Ireland (1.965 ppm). However, in 2020, easterly winds advected low atmospheric $CH_4$ concentrations (1.98 ppm). These winds originated from the less populated and more agriculturally used land areas of Schleswig-Holstein, or even from the Baltic Sea. The mean wind speed of these easterly winds was 5 m s$^{-1}$, and the distance from Helgoland to the Baltic is about 200 km. Thus, the wind covered this distance within eleven hours and the air mass we were measuring could

have come again from an open sea area, this time the Baltic Sea. This air mass origin is supported by the NOAA



modeling, as air masses observed in our study area at the end of 1 September 2020 originated from Schleswig-Holstein and the Baltic Sea (Fig. S6).

### 4.5    Conclusions

In our study we compared different methods to calculate the diffusive $CH_4$ fluxes with in-situ data and data from land-based meteorological stations. The usage of in-situ wind data (at high temporal resolution) was most important, while the usage of the in-situ atmospheric concentration data showed no large difference to fluxes obtained using in-situ data. When extrapolating from the measured data and from the real study area to a larger area (i.e., Helgoland Bay) it was important to use the arithmetic average and not the median value. Most natural

data are skewed towards the lower values, and using the median of these datasets would result in an underestimation of diffusive $CH_4$ flux. The area-weighted extrapolation, however, is recommended, as it yielded the most realistic results with smallest variability.

We observed large variability in our data sets, which was not due to methodological constraints but reflects the high natural variability of the study area. Thus, it is debatable if it is reasonable to average over a

heterogeneous area such as the Helgoland Bay. An improvement of flux estimates could be achieved by covering the whole area with a systematic zigzag track.

Hot spots of $CH_4$ emissions were the tidal flats at low tide. Their $CH_4$ emissions resulted in locally elevated atmospheric $CH_4$ concentrations. However, in shallow water and rough sea, the coastal North Sea was undersaturated with $CH_4$ and acted as $CH_4$ sink. Overall, the diffusive $CH_4$ flux into the atmosphere accounted

for increased atmospheric $CH_4$ concentrations only at low wind speeds. Atmospheric advection was the main driver for low $CH_4$ concentrations (when coming from the sea) or high $CH_4$ concentrations (when coming from the mainland).

With our comprehensive study we revealed a complex relationship between dissolved $CH_4$ concentrations, $CH_4$ fluxes to the atmosphere and atmospheric $CH_4$ concentrations in shallow coastal water

areas.

**Acknowledgements:** We thank the crews of our research vessels (Littorina, Ludwig Prandtl, Mya II, and Uthörn) for their support and patience. Special thanks are due to Norbert Anselm and Lea Happel for their

unfailing support with our data work. Many thanks also to Jens Greinert & Tim Weiss as well as Uta Ködel for providing atmospheric $CH_4$ measurements on the RV Littorina and on RV Ludwig Prandtl, Mario Esposito and Felix Geissler for leading the Littorina cruises in 2029 and 2020.
This study is part of the Helmholtz program Changing Earth, subtopic 4.1: "Fluxes and transformation of energy and matter in and across compartments". We acknowledge funding from the Helmholtz Association in the

framework of the Helmholtz funded observation system MOSES (Modular Observation Solutions for Earth Systems).

**Author contribution:** IB, HB, GF and PF planned and participated in the cruises. Data were prepared by IB and PF. IB prepared the manuscript with contributions from all co-authors.




**Competing interests:** NB is member of the editorial board of the journal Biogeosciences. The authors have no other competing interests to declare.

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
