# Peer review of "Influence of wind strength and direction on diffusive methane fluxes and atmospheric methane concentrations above the North Sea"

_EGUsphere, 2023_

## Author Response (AR1)

Referee 1

The current manuscript presents a dataset of diffusive CH4 fluxes, as well as measurements of dissolved and atmospheric methane concentrations in the North Sea. The authors study the important topic of sea-atmosphere exchange of CH4 in coastal zones, which is associated with high uncertainty nowadays. They measure with different equipment, on 3 research vessels and use different approaches to calculate diffusive fluxes to study the influence of different variables and measurements on the calculations. Thus, this dataset of methane measurements contributes to improve the knowledge about the dynamic of this greenhouse gas in coastal areas. Furthermore, this study adds more data to the overall collection of methane measurements.
I think that this paper achieves the aims of the journal, but I have some corrections and suggestions to render the work more attractive to readers.

Review
-L.115-117: Replace "CH4" by "CH4", with the subscript.
*Corrected*

-L.193: What is the model of the Microportable Greenhouse Gas Analyzer (LosGatos)? Is it the same as the one used in Stern-5: LosGatos GGA-911?
*Corrected*

-L.215: The authors have used one parameterisation for the k calculation. Since this work focuses on studying and trying to reduce the uncertainty associated with the calculation of diffusive fluxes of methane to the atmosphere and they indicate that the determination of k is crucial to calculate the flux, why do the authors use a single parameterization instead of considering using another parameterisation of k to compare the results?
*We agree that a discussion on the parametrization of k-600 might be necessary and will insert the following to the discussion:*

*The calculation of the diffusive sea-air flux depends very much on the parametrization of k. The most frequently used formula is the one from Wanninkhof 2014. For comparison we applied this formula to our data set from 2020; and average flux was very similar $153 \pm 441$ $\mu mol\ m^{-2}\ d^{-1}$ according to Wanninkhof versus $159 \pm 444\ \mu mol\ m^{-2}\ d^{-1}$ according to Nightingale. Both parametrizations should* provide good estimates for most insoluble gases at intermediate wind speed ranges (3-15 m s–1). Our wind data ranged from 1 – 11 m/sec.
*The study of [Ho, 2018 #3339] concludes that if the mean depth of the water body is greater than 10 m, an ocean wind speed/gas exchange parameterization could be used in such environments. The mean water depth in our study area $19 \pm 12$ m in 2019 and $17 \pm 13$ m in 2020. We therefore believe that the parametrization of Nightingale is appropriate for our study area. However, it should be kept in mind that also this parametrization holds an uncertainty of 19%. Other factors influencing the parametrization of k are rain (which did not occur during our cruises), water-side convection, and a biological surfactant suppression term [Gutiérrez-Loza, 2021 #3319]. During summer, convection and surfactants seemed to act as competing mechanisms controlling the flux. Convective processes slightly enhanced the downward flux, while surfactants tended to suppress it [Gutiérrez-Loza, 2021 #3319].*

- L.228-2235: The authors used three combinations of data sets, using different wind and atmospheric CH4 data to calculate the k. There is a possible fourth combination, using in situ

dissolved CH4 and atmospheric CH4 data, and the wind speed data obtained from DWD. why has it not been used?

*Yes, there is a fourth combination possible, but the idea was to start with all in situ data and then stepwise replace them by data from weather stations. The last combination with in situ atmospheric methane and wind from DWD, thus seemed to be not realistic.*

- L. 265-276: Since the authors used different greenhouse gas analyzers for the different vessels and cruises (Picarro G2301, Licor LI-8100A and LosGatos GGA-911), which sensor is used as a lead sensor?

*The lead sensor was the Picarro G2301 on the Littorina, more detailed informations are given in additional information in the data publication, Pangaea, as outlined in L-280*

-L.240-247: Which diffusive flux (flux-1, flux-2 and flux-3) do the authors use for the calculations of the area-weighted diffusive flux? Include it in the paragraph.

*The flux-1 data were used, as is now mentioned in the text*

-L.280: Remove or replace "##".

*The now published doi for Sternfahrt 3 is now given*

-L.286-288: Include the standard deviation or range, as it has been done for atmospheric CH4 concentrations, since wind speed is one of the main variables studied in this work.

*This information is now given for both cruises.*

-L.304-311: The diffuse fluxes for Stern-3 calculated with the in-situ dataset (flux-1) are presented. Why are flux-2 and flux-3 not shown? Moreover, are the CH4 fluxes presented for Stern-5 (lines 333-337) also those calculated with the approach of flux-1? If it is so, why are flux-2 and flux-3 not shown as well?

*We have now inserted the following information for both cruises: "flux-2 and flux-3 data are shown in Table 2 and described in section 3.3)." The sections 3.1 and 3.2 focuses more on the description of the environment, and section 3.3 then focusses more on the "methodological aspects".*

-L.365-387: The results of the different approaches used to calculate the diffusive fluxes (flux-1, flux-2 and flux-3) are shown in subsection 3.3 of the results. Since the authors compare the diffuse fluxes in this subsection, they should include the diffusive CH4 fluxes (flux-1) described in subsections 3.1 and 3.2 in subsection 3.3 to compile the data in the same subsection.

*The data of flux-1 are also shown in Table 2. As we used different regions for the wind data, also the flux-1 data had to be splitted into these regions. Therefore, we prefer to discuss the differences between the calculations in section 3.3, and in the sections 3.1 and 3.2 focus on the environmental description.*

-L. 336-337: It is stated that: "The data for dissolved and atmospheric CH4 and the diffusive CH4 flux for the individual days are shown in Figure S2, analogous to Fig. 4.". However, the atmospheric CH4 for the individual days is also shown in Figure 8. The information is repeated. Why do the authors show atmospheric CH4 concentration per day of sampling in Figure 8 but show concentrations of dissolved CH4 and diffusive fluxes for the whole study period (instead of per day) in Figure 6, instead of using Figure S2? Use Figure 8 and remove the atmospheric CH4 concentration from Figure S2 or include Figure S2 instead of Figures 6 and 8.

*As the cruise in 2020 lasted 4 days, we thought that displaying each parameter for each day would be too much of information. The reviewer is correct that there is a doubling of information for the atmospheric CH4. We thus will remove the atmospheric CH4 from figure S2.*

-L. 438 and 444: The parentheses are not correctly placed and/or another one is needed.
*Corrected*

Figures
Figures 1 and 2:
There are some zones commented on the results/discussion section, such as Islands of Scharhörn and Neuwerk, which are not shown in the figures. Include the location of these areas on the maps.
*corrected*

Figure 2:
"wy" and "bü" are included in the figure but have not been included in the figure caption. Include the meaning of the acronyms in the caption.
*Corrected*

Figure 4:
The third figure on the right, "CH4 dry [ppm]" has a different font size.
*Will be corrected*

Figure 6:
Is the first day of sampling of Stern-5 30 August or 31 August? In the manuscript the authors have written 31 August but in the figure caption, 30 August.
*Corrected to 31 August*

Figure 8:
There is an error in the figure caption: The date of the sampling on Figure 8 top left is 31 August, not 30 August.
*corrected*

Tables
Table 3:
The table caption indicates "comparison of three approaches", but in the table, it shows the mean and standard deviation for the two approaches described in the subsection 3.4 of results: the first, which is described in the Method section, and the second, described in lines 422-423. Also, it includes the median and range of the diffusive fluxes. Is this median and range calculated with the first approach?
*Changed to*
*Table 3. Comparison of three approaches to calculate the total diffusive flux from Helgoland Bay with an area of 3.78 x 109 m2 based on the median, mean or area-weighted diffusive flux.*

**Reviewer comment on „Influence of wind strength and direction on diffusive methane fluxes and atmospheric methane concentrations above the North Sea" by Bussmann et al.**

General:
The paper addresses flux estimations of methane from the southern North Sea based on in situ water concentrations, in situ atmospheric methane mole fractions and in situ wind speed data, and compares the results to flux estimates based on the same air-sea exchange parameterization (Nightingale et al, 2000), but using either station -derived monthly mean data for the atmospheric mole fractions, or the same monthly mean data plus wind-speed data from three different coastal meteorological stations. The authors use the comparison of the different results, as well as the effect of using the mean and median value of their data for providing average integrated fluxes, to assess the main drivers of uncertainty, as well as to give recommendations.

While the data, if traceable (see below) would be of high value as a contribution to the data on coastal methane concentrations and fluxes, I found the approach not really suitable and state-of-the-art to answer the questions addressed, and therefore suggest major revision or rewrite of the paper with an altered focus.

In principle, the authors follow a classical approach using a wind-speed parameterized air-sea flux calculation based (the kinetic term) in connection to the disequilibrium between sea surface methane concentrations and equilibrium concentration with the overlying atmosphere (Delta C – the thermodynamic term).

The authors do not address uncertainties in using the chosen parameterization itself, quoting e.g. Ho et al. 2018, who suggests that usage of such an approach might be valid for water depth larger than 10m, despite the fact that some of the waters investigated are shallower. Given the aforementioned scope of the paper, I miss a discussion of the spread in k-parameterizations and issues for an area like the one treated, as well as some background information on additional processes (e.g. Gutiérrez-Loza et al., 2021, J. Mar Systems 222);

*We agree that a discussion on the parametrization of k-600 might be necessary and will insert the following to the discussion:*

*The calculation of the diffusive sea-air flux depends very much on the parametrization of k. The most frequently used formula is the one from Wanninkhof 2014. For comparison we applied this formula to our data set from 2020; and average flux was very similar $153 \pm 441$ $\mu mol\ m^{-2}\ d^{-1}$ according to Wanninkhof versus $159 \pm 444\ \mu mol\ m^{-2}\ d^{-1}$ according to Nightingale. Both parametrizations should provide good estimates for most insoluble gases at intermediate wind speed ranges (3-15 m s–1). Our wind data ranged from 1 – 11 m/sec. The study of [Ho, 2018 #3339] concludes that if the mean depth of the water body is greater than 10 m, an ocean wind speed/gas exchange parameterization could be used in such environments. The mean water depth in our study area $19 \pm 12$ m in 2019 and $17 \pm 13$ m in 2020. We therefore believe that the parametrization of Nightingale is appropriate for our study area. However, it should be kept in mind that also this parametrization holds an uncertainty of 19%. Other factors influencing the parametrization of k are rain (which did not occur during our cruises), water-side convection, and a biological surfactant suppression term [Gutiérrez-Loza, 2021 #3319]. During summer, convection and surfactants seemed to act as competing mechanisms controlling the flux. Convective processes slightly enhanced the downward flux, while surfactants tended to suppress it [Gutiérrez-Loza, 2021 #3319].*

A major issues I have with the paper is the lack of information on the method used for the measurement of CH4 concentrations, despite the fact that data using the method have been already published in Bussmann et al., 2021b. To my understanding, the group does not use a circulating air-sea exchange equilibrator system attached to a CEAS sensor (e.g. Pfeill et al., 2012 for CO2, Gülzow et al., 2011), but relies on the partial stripping of CH4 from the water by a CH4-free stream of carrier gas. This method has been tried in the past but produces major issues as it is way-dependent (i.e. dependent on the kinetics) of the equilibration system, requiring very stable operation conditions, and are potentially dependent on temperature, salinity, and possible contamination of the membrane system. Bussmann et al. try to overcome this, to my understanding, by taking at least hourly discrete samples and measuring those by a headspace method. Firstly, I miss the information of this procedure by showing a plot of "relative ppm" against discrete measurements over time and/or concentration. Also, in particular for low concentrations, it has been shown that discrete samples have their limits as well, usually with purge and trap systems working better than headspace approaches for low concentrations (e.g. Wilson et al., 2018). This is e.g. important in connection to the finding of undersaturated waters in an area where one would not expect those. In other words, there is a lack of a) information to judge the derivation of the concentration, including a plot showing the basis for the "relative ppm to concentration conversion b.) a representation of the concentration findings in a histogram plot like Figs S4&5), as well as a robust estimate of the error for the method.

*The description of the method may have been too short and we will elaborate the method and its limitation.*

*Two example plots for the calibration of the GGA (ppm-GGA versus nmol/L from water samples) and the details of all calibrations will be given in the supplements. Also, two histograms for the methane concentrations of the two cruises will be added to the supplements.*

*As outlined in the data management section of the ms, we had intercalibration stations where all ships stayed close together sampling the same water for 30 min. From these intercalibrations stations we derived a standard error for the method of 3.6% (n = 7). This information will be added to the main text.*

*For a different cruise, but with the same instrumental setting, aerated freshwater with an equilibrium concentration of 2.9 nM was measured. The GGA showed concentrations of 2.3 ± 0.3 nM. Thus, we are confident in using a reliable instrumental set-up.*

The other side of the "DeltaC- part" investigated here is the choice of the airborne methane partial pressure. Not surprisingly, given that large oversaturations drive the area integrated ASE-flux, the effect is minor. So the merit to do these concentration measurements is apparently more related to the question whether the fluxes from the area (in particular the Wadden Sea) have a measurable imprint on the methane concentration of the marine boundary layer. This part is straightforward, though I would think that a discussion in the framework of methane budgeting by inverse atmospheric modelling would be beneficial here.
*Thank you for the suggestion to use inverse atmospheric modelling, which we will aim for it in the near future.*

A major difference between the different calculated fluxes is caused by the choice of the wind speed data source, and the paper recommends to use in situ wind data. There are some issues with the approach. First, state of the art work without in situ measurements would actually

NOT use the data from these individual stations, but rather a modelled wind product like the COSMO CLM fields, potentially comparing the results of different wind products. So the comparison made is against something which cannot be considered state of the art. Moreover, the argument that the heterogeneity in the wind forcing has to be taken into account has to be discussed in connection to the wind speed parameterization used. This is very nicely discussed in the "old Wanninkhof 92" paper, who at that time came up with two parameterizations for short term and long term wind fields to account for the wind speed distribution and the quadratic response. So the parameterization and the time resolution of the wind speed have to "match" . While Rik Wanninkhof urges to use the parameterization in W 2014 instead of 2009, the argumentation on the dependency between any parameterization and the time resolution of the wind speed data remains valid.

*In our study we compared the application of in-situ wind versus data from nearby meteorological stations (with a hourly resolution). Long term data as discussed in Wanninkhof rely on yearly wind averages, this may be appropriate for the open ocean, but probably not for coastal seas. Many other studies also use wind-data from nearby metrological stations, and our point was to compare these two applications. The application of a modelled wind, would also have been an option. However, (wind) models also have their limits as they only approximate reality. Thus, our aim was to compare the application of two observational datasets, from different locations and different resolution, as we did with the comparison of observational atmospheric methane concentrations.*
*New methods are available to overcome spatial and temporal restrictions of observed data sets mostly for CO2 [Bittig, 2024 #3405]. Yes, it would be interesting to apply these new calculations to our data set.*
*The following sentence is now added to the conclusions:*
*New statistical methods are now available to overcome spatial and temporal restrictions of observed data sets mostly for $CO_2$ [Bittig, 2024 #3405] and their application might give new insights.*

Lastly, the authors investigate the use of mean vs. median of the measurements for area-integrated flux estimates, and also acknowledge that an evenly coverage of the area of interest (zigzagging) would be beneficial. I think that the authors make a valid point here that due to the linear relation of fluxes to DeltaC, in fact small areas of high flux have a large impact on the area-integrated flux, and the mean rather than the median would be the choice, as long as the data do not have a bias (i.e. more data on high flux areas). In that regard, it is unfortunate that the supplemental figure S1 was not readable in the download. The figure should be modified in a way that it is represented where data exists and for the extrapolation of which area they have been used. This would allow to easily judge whether the data are locally representative to some extent.
*A new map is now provided with better quality as Figure S2, the name of the area and its key number as provided from the German Statistical Service.*
*To circumvent the influence of small areas with high flux we applied the area weighted flux, as outlined in section 2.4.*

Also, the authors miss the opportunity to acknowledge, or better use, other potential extrapolation schemes. For instance, Borges et al. made the point that distance to the coast turned out to be a good proxy for CH4 concentrations in his study. In this study here by Bussmann et al., the authors themselves state that the highest concentration were encountered in towards the Wadden Sea, and mention a potential impact of the tides. However, this is not considered at all in their discussion of the options for interpolation.

*We will amend the discussion for the aspect "distance to the coast", however no such correlation could be found in our dataset*
*Another aspect of elevated CH₄ concentrations, is the distance to the coast, as described in several studies [Sierra, 2020 #3144] [Thomas, 2012 #3404]. However, no such correlation was observed in this study; probably due to the a highly diverse coast (Wadden Sea, sandy beaches, estuaries) superimposed by tidal cycles.*

Minor and more tailored comments (sometimes redundant to above):

- Line 297 : is is very hard to understand why data should have been undersaturated anywhere in this area; which again gives rise to the need for a detailed discussion of the accuracy and reproducibility of the method;
- *As outlined above the method is now described in more detail. In section 4.3 we discuss that at this shallow station with strong winds, the water column has been depleted of methane.*

- Split of days: the rationale for this split is the shift in the wind field and air masses; so when did that happen really?; the day time cut looks arbitrarily; original data against time (best for all 3 DWD stations and the in situ wind data in the supplement would allow to better judge when the shift in air massed occurred (and also would be beneficial for the discussion of the winds to be used)
- *Wind data are available for the whole duration of the cruise, however dissolved methane and flux data only when the ships were cruising during day time. Thus, when exactly the wind regime shifted seems not so important.*

- The always lower median than mean points to a majority of low values; data histogram would be helpful; the accuracy of the airborne CH4 might be only important for the lower ones. Add info as in Fig S4 and S5 for the fluxes but for concentrations.
- *Histograms for the concentration of dissolved methane are now provided in the supplements.*

- The account of the different frequency contributions mentioned in connection to Fig S4 and S5 are in the text "converted" to percentages, but seem not to add up to 100%. But maybe I overlook something here.
- *The wording was misleading here, and is corrected to add up to 100%*

- Chapter 3.5: here, an interaction with a full grown inverse atmospheric model could have shed some light, in particular on how much flux would be needed to actually see the fluxes from the Wadden Sea as elevated methane.
- *Yes, the application of an atmospheric model could have given new insights, but it would also need expertise to do so. In future, this certainly could be done.*

- Wind height correction: give reference for equation (1) for the u10 conversion.
- *The reference is now given in the revised Ms, (Touma 1977)*

Supplement:
Hydrographic parameters: please be more specific on the sensor equipment; there are various types of Aandera optodes and Meinsberg pH electrodes etc.
*As far as possible, these informations are now given.*

Fis S3 caption: change "within-situ" to "with in-situ"

*corrected*